# Is There a Therapeutic Benefit of Axillary Surgery in Non-Metastatic Breast Cancer? A SEER Cohort Database Study

**DOI:** 10.3390/jcm14176314

**Published:** 2025-09-06

**Authors:** Jonathan Sabah, Alexis Marouk, Sébastien Molière, Massimo Lodi

**Affiliations:** 1Breast Reconstructive and Plastic Surgery, Louis Pasteur Hospital, 68000 Colmar, France; jonathan.sabah@chru-strasbourg.fr; 2Assistance Publique—Hôpitaux de Paris (APHP), 75610 Paris, France; 3Department of Women’s Imaging, University Hospitals of Strasbourg, 67200 Strasbourg, France; 4Institute of Genetics and Molecular and Cellular Biology (IGBMC), CNRS UMR7104 INSERM U964 University of Strasbourg, 1 rue Laurent Fries, 67400 Illkirch-Graffenstaden, France

**Keywords:** axillary lymph node dissection, sentinel lymph node biopsy, breast cancer, survival analysis, cohort study

## Abstract

**Background.** Axillary lymph node biopsy (ALND) has traditionally been considered the gold standard for axillary staging and treatment in clinically node-positive breast cancer patients. However, in patients with nodal disease, the therapeutic benefit of ALND is uncertain. This study, based on a large cohort, aims to evaluate breast cancer-specific survival depending on the extent of axillary surgery in non-metastatic breast cancer using real-world data from the Surveillance, Epidemiology, and End Results (SEER) database. **Methods.** This retrospective cohort study comprised 825,240 patients diagnosed with breast cancer between 2000 and 2020. **Results.** ALND was associated with a worse survival outcome in pN0 and pN1 populations (respectively, hazard ratio [HR] 1.16; 95% confidence interval [CI] 1.12–1.2; *p* < 0.001 and HR 1.38; 95%CI 1.3–1.46; *p* < 0.001). In pN2 and pN3 populations, there was ~4.3% relative reduction in the hazard of breast cancer-related death for each additional node removed; and higher positive-to-removed lymph node ratio was associated with worse prognosis (HR 3.450; 95%CI 2.99–3.98; *p* < 0.001). **Conclusions.** SLNB is associated with significantly better specific survival compared to ALND in negative/low axillary involvement, in higher axillary involvement categories extensive axillary surgery was associated with better prognosis.

## 1. Introduction

Current surgical management of non-metastatic invasive breast cancer surgery includes complete surgical excision of the primary tumor and locoregional staging of axillary lymph nodes. Axillary staging can be clinical, radiological, and surgical. Surgical staging includes axillary lymph node dissection (ALND) and sentinel lymph node biopsy (SLNB). ALND, a more extensive procedure, has traditionally been considered the gold standard for axillary staging and treatment in clinically node-positive breast cancer patients [1,2]. Indeed, the extent of metastatic lymph node involvement is the major prognostic factor in breast cancer and can influence subsequent treatment decisions for chemotherapy, radiotherapy, targeted, and endocrine therapies [2].

For clinically node-negative patients, sentinel lymph node biopsy (SLNB) is the preferred technique today and can be performed for T1 and T2 tumors. Over recent decades, the use of SLNB has risen significantly, largely replacing ALND in clinical practice for patients with breast cancer. This shift reflects evidence from multiple randomized trials, such as the Milan [3], NSABP-B32 [4], ALMANAC [5], and ACOSOG Z0011 trials [6]. SLNB can effectively identify metastatic involvement while reducing the complications associated with more extensive lymph node removal—such as pain, breast cancer-related lymphedema, and postoperative complications [7] with non-inferior overall survival [1]. This trend is also observed for initially node-positive (cN1) breast cancer patients who convert to node-negative after neoadjuvant chemotherapy [8,9,10,11].

Regardless of the benefits of precise axillary staging, which are necessary to tailor the adjuvant treatment and therefore improve outcomes, it has been hypothesized that the surgical removal of metastatic lymph nodes also has a direct therapeutic effect by reducing tumor burden. However, no studies have conclusively demonstrated that ALND offers a therapeutic benefit in terms of improving survival compared to SLNB [12,13]. According to different meta-analyses of randomized trials, there seems to be no significant benefit in survival for patients undergoing ALND compared to those undergoing SLNB [14,15]. This study, based on a large cohort, aims to evaluate breast cancer-specific survival depending on the extent of axillary surgery in non-metastatic breast cancer patients using real-world data from the Surveillance, Epidemiology, and End Results (SEER) database.

## 2. Materials and Methods

### 2.1. Study Design and Data Source

This retrospective cohort study utilized data from the SEER database (SEER Research Data, 17 Registries, November 2022 Sub, 2000–2022), which collects comprehensive cancer incidence and survival data from population-based cancer registries across the United States, including patients diagnosed with breast cancer between 2000 and 2020. This study was conducted following the Strengthening the Reporting of Observational Studies in Epidemiology (STROBE) guidelines [16].

### 2.2. Patient Selection (Figure 1)

Inclusion criteria were patients aged 20 years or older with a primary diagnosis of epithelial invasive non-metastatic breast cancer who underwent upfront axillary surgery. Non-inclusion criteria were:-Ductal carcinoma in situ or microinvasive cancer only (Tis and Tmi category, stage 0).-Patients with breast cancer staged as M1.-T3 or T4 tumors (due to the high false-negative rates of SLNB).-Non-epithelial tumors: sarcomas, phyllodes tumors, etc.-Neoadjuvant systemic treatment.-No surgical procedure or missing data on axillary surgery (no lymph nodes removed).-Missing data on survival, positive, or removed lymph nodes.

**Figure 1 jcm-14-06314-f001:**
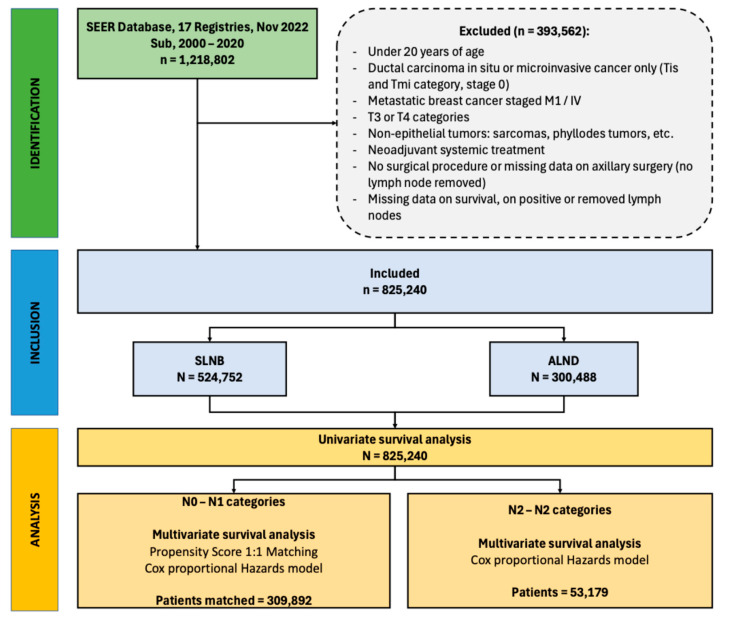
Patient selection flowchart.

### 2.3. Definition of Axillary Surgery

The exact type of axillary surgical procedure is not explicitly reported in the SEER Database (SLNB or ALND), nor whether it was upfront ALND or completion after initial SLNB. To infer the type of axillary surgery, we developed a two-step procedure, including a rule-based classifier followed by a Bayesian logistic regression model. The first step consisted of classifying SLNB patients with 1–3 removed lymph nodes, or ≤4 removed lymph nodes and N0 category. Patients with ≥10 nodes removed, or ≥7 nodes and initially node-positive, or classified as N2/N3 categories were assigned to the ALND group. The remaining unknown axillary procedures were assigned to the ALND and SLNB groups according to a Bayesian logistic regression model using a Bernoulli function. This model was developed using the R’s brms package (version 4.4, 2025, R core team, Austria) and was calibrated using informative priors derived from clinical assumptions as follows:-The number of removed lymph nodes (normal distribution, high weight, high confidence): the more lymph nodes that are removed, the more likely an ALND was performed.-Year of diagnosis (normal distribution, low weight, low confidence): in 2000, there were more ALND compared to SLNB than in 2020.-N category (normal distribution, high weight [and higher for N2 and N3], intermediate): if a patient had high axillary involvement, the more likely it is that an ALND was performed.

Then, we trained the model on the previously categorized (ALND and SLNB) subgroups to probabilistically predict the likelihood of ALND in the remaining unknown axillary procedures. Patients with a posterior probability > 0.50 were reclassified as ALND, and those with a posterior probability < 0.50 as SLNB. We considered this approach to be more robust compared to considering that SLNB was performed if 4 or fewer lymph nodes were removed, while ALND involved the removal of at least 10 lymph nodes, because there would not be an intermediate category of unknown axillary surgery (as in clinical practice). However, this approach did not allow us to discriminate upfront ALND and SLNB followed by ALND.

### 2.4. Statistical Analysis

All statistical analyses were conducted using R software (version 4.4, 2025, R core team, Austria) and the following packages: tidyverse, MatchIt, cobalt, survival, bmrs and survminer. Statistical significance was defined as a two-sided *p*-value < 0.05. Descriptive results contained missing values. Multiple imputation using chained equations (MICE) was performed using the predictive mean matching method to handle missing data and to allow for robust subsequent analyses. This imputation method accounted for the uncertainty of the missing data.

Propensity score matching was used to minimize selection and indication bias, which occur when the groups being compared differ in ways that affect the outcome and when treatment assignment is based on disease severity. This method balanced baseline characteristics between patients undergoing SLNB and ALND, ensuring a fair comparison of breast cancer-related survival outcomes between the two groups. Matching was performed using a 1:1 nearest neighbor algorithm without replacement, based on variables including age, year of diagnosis, tumor subtype, tumor grade, estrogen receptor status, progesterone receptor status, HER2 status, T stage, N stage, systemic treatment, breast surgery type, and radiotherapy. Non-redundant items were checked in both the propensity score model and the MatchIt algorithm to ensure robust matching. We applied a caliper of 0.2 times the standard deviation of the logit of the propensity score, as recommended in the literature [17]. To ensure the quality of matching and balance between groups, we utilized love plots to visualize standardized mean differences (SMDs), with acceptable values set at SMD ≤ 0.1. Common support was verified to ensure proper overlap in propensity scores between the SLNB and ALND groups.

Kaplan–Meier survival curves were generated to compare overall survival between the SLNB and ALND groups, and the log-rank test was used to assess differences. Survival analysis was conducted based on matched populations with the propensity scores. Univariate and multivariable Cox proportional hazards models were employed to estimate hazard ratios (HRs) and 95% confidence intervals (CIs) for the association between the type of axillary surgery and overall survival, adjusting for potential confounders. Robust estimators of variance were used for matched data analysis.

For pN2 and pN3 patients, there was no possible comparison of SLNB and ALND, as there were no SLNBs in this population. Therefore, we investigated whether the number of examined lymph nodes impacted specific survival. In addition, to avoid collinearity of the Cox model, we calculated the positive-to-removed lymph node ratio ($ratio=\frac{n_{positive\;nodes}}{n_{removed\;nodes}}$) and also evaluated its impact on specific survival, based on the hypothesis that a higher ratio was associated with an higher probability of “insufficient axillary clearance”. The positive-to-removed lymph node ratio has been positively evaluated in node-positive breast cancer to predict survival [18,19].

### 2.5. Ethical Considerations

The use of de-identified patient data from the SEER database was deemed exempt from institutional review board (IRB) approval.

## 3. Results

### 3.1. Patient, Tumor, and Treatment Characteristics

The initial cohort comprised 1,218,802 patients, with 825,240 patients included in the analysis after exclusions (N = 393,562). Among the total cohort, 63.6% had SLNB and 36.4% had ALND. Patient, tumor, and treatment characteristics are described in Table 1. For this survival analysis, we matched 250,592 patients with N0 and 59,300 patients with N1 statuses. For N2+ categories (53,179 patients), the propensity-score matching was not possible as no SLNB were performed, therefore we analyzed these categories separately.

### 3.2. Total Cohort Survival Analysis

Survival analysis included 825,240 patients with 68,689 breast cancer-related deaths recorded. The median follow-up time until cancer-related death or loss of follow-up was 91 months (7.58 years). Univariate survival analysis showed better survival in the SLNB group (*p* < 0.001, Appendix A
Figure A1). The results of the Cox models—adjusted on the year of diagnosis, age, T category, tumor grade, immunohistochemistry (ER, PR, HER2), breast-conserving surgery, radiotherapy, adjuvant systemic treatment, time between diagnosis and treatment, and number of positive lymph nodes—showed that ALND was associated with a worse survival outcome (HR 1.96; 95%CI 1.91–2.00, *p* < 0.001), and the number of removed lymph nodes was proportional to higher specific mortality (HR 1.0033; 95%CI 1.0022–1.0045, *p* < 0.001).

### 3.3. pN0 and pN1 Survival Analyses

In the propensity score-matched populations, ALND showed a statistically significant increase in the risk of breast cancer-related death compared to SLNB, both for pN0 and pN1 categories (*p* < 0.001, see Figure 2). Hazard ratios of ALND are reported in Table 2.

### 3.4. pN2 and pN3 Survival Analysis

In the pN2 and pN3 populations, the multivariable Cox models were adjusted on the year of diagnosis, age, T category, tumor grade, immunohistochemistry (ER, PR, HER2), breast-conserving surgery, radiotherapy, adjuvant systemic treatment, time between diagnosis and treatment, and the number of positive lymph nodes. Both models showed a favorable impact of extended axillary surgery on prognosis (Table 3). Indeed, the number of lymph nodes removed was significantly associated with improved survival (HR 0.957; 95%CI 0.955–0.960; *p* < 0.001), indicating a ~4.3% relative reduction in the hazard of breast cancer-related death for each additional node removed. The higher positive-to-removed lymph node ratio was associated with worse prognosis (HR 3.450; 95%CI 2.99–3.98; *p* < 0.001).

## 4. Discussion

This large population-based analysis confirms previous findings on the impact of axillary surgery in patients with non-metastatic non-post-neoadjuvant breast cancer on specific survival. Indeed, in clinically and pathologically node-negative (c/pN0) or limited node-positive (pN1) patients, these results show that SLNB is associated with a significantly better specific survival compared to ALND, even after rigorous adjustment. Conversely, for patients with advanced nodal disease (pN2 and pN3), these results indicate that the extent of axillary dissection may have a prognostic role. Specifically, a greater number of lymph nodes removed was independently associated with improved breast cancer-specific survival, while a higher positive-to-removed lymph node ratio was associated with worse outcomes.

These findings support the evidence that a personalized approach is critical when planning axillary surgery for early breast cancer, as the therapeutic benefit of ALND in low nodal involvement is uncertain, meanwhile in patients with extensive nodal disease, more complete surgical clearance may reduce residual tumor burden and improve survival. This aligns with the hypothesis that in the latter situation axillary surgery may retain a therapeutic component beyond staging alone; however, it must be noted that differences in survival should not be confounded with a guaranteed therapeutic effect of ALND, as adjuvant therapies (both systemic and radiotherapy) are determined by the extent of axillary surgery.

For clinically N0 and some N1 patients, the standard treatment today is SLNB in most countries worldwide. The results of this study align with the findings of level 1 evidence studies [3,4,5,6,20,21], which allow ALND to be avoided for patients with up to two involved lymph nodes [22], or subsequent ALND after positive SLNB [21]. Therefore, it seems that in this low/null axillary involvement context, the benefit of axillary surgery lies in correct nodal staging and reducing the risk of undertreatment of clinical and radiological staging only.

Conversely, for higher axillary involvement (palpable node-positive disease, and clinically N2 and N3 categories), the standard of care is ALND when up-front surgery is planned [2]. Similar to our findings, a retrospective cohort analysis of the National Cancer Data Base between 2004 and 2013 by Park et al. included 22,156 patients with clinically N2/N3 categories—however, including metastatic patients in the analysis—and found that ALND compared to SLNB improved survival [23]. On the other hand, a study published by Ebner et al. of a multicenter German database of 9625 women found that removing ≥ 10 lymph nodes did not improve survival in all node-positive patients [24]; however, there was no specific analysis of the higher N2 and N3 categories. In this higher axillary involvement population, patients may undergo neoadjuvant treatment aiming for nodal clinical and radiological complete response, permitting the omission of ALND for Targeted Axillary Dissection [22,25]. Nevertheless, nodal complete response in the most frequent subtype (hormone-positive, HER2-negative) is infrequent (~20%), resulting in low rates of surgical de-escalation after neoadjuvant treatment [26,27]. Differences in survival between SLNB and ALND for node-positive patients can also be explained by differences in adjuvant radiotherapy. Indeed, those patients undergoing SLNB only are more likely to receive axillary radiotherapy [1,26], which can also has been proven to effectively reduce tumor burden and to be non-inferior to complementary ALND [21].

Determining the actual quantity of positive nodes may inform adjuvant treatments, specifically in relation to the risk of SLNB understaging and subsequent undertreatment. This risk has been investigated and depends also on clinical N staging. It has been shown that clinical and pathological N categories are concordant approximatively 80% of cases, and the majority of discordances are in favor of upstaging with pathological examination (18.5%) [28]. For clinical N0 patients—which includes the SLNB population—upstaging was found to be 17.9%, however only 2.9% for pN2/pN3 categories [28]. Today, axillary ultrasound ± biopsy is considered the “best” radiological procedure to preoperatively assess lymph node involvement, however the risk of understaging is still high, even in specialized centers [29]. Conversely, patients with clinically suspected lymph node involvement (cN1–3) that are downstaged into pN0 ranges between 19 and 27.5% [28,30], underlying the importance of preoperative axillary cytology/biopsy in order to avoid unnecessary ALND.

Still, the decision regarding adjuvant treatments may be less impacted than expected. Indeed, in menopausal women the number of positive lymph nodes influences chemotherapy decisions, alongside the 21-gene assay recurrence score [31]. Nevertheless, research conducted on 1786 post-menopausal women diagnosed with cT1-2N0, HR+/HER2− breast cancer indicated that the implementation of subsequent ALND contributes only a 1% increase (from 12% to 13%) in chemotherapy recommendations based on the RxPONDER criteria [32].

Similarly, cyclin-dependent kinases (CDK) 4/6 inhibitors may also be indicated based on lymph node involvement. The first available treatment with abemaciclib [33] required the presence of at least four positive lymph nodes, or one to three positive lymph nodes accompanied by two of the following criteria: grade III tumor or size ≥ 5 cm. This requirement sometimes resulted in the completion of ALND following a positive SLNB to satisfy the prescription criteria. Nonetheless, it has been demonstrated that the number of patients needed to treat with ALND for one woman to benefit from abemaciclib is 139 [34]—and a 3.3% benefit in another study [35]—thus discouraging this method. Furthermore, with the recent publication of the NATALEE trial findings, the necessity for a sufficient number of positive nodes will no longer be a criterion for adjuvant CDK4/6 inhibitors [36]. Nevertheless, the consideration of de-escalation in favor of SLNB incurs an escalation in radiotherapy [25,37].

It must also be noted that it is, in part, because of these adjuvant treatments that axillary recurrences are extremely low (0.9% and 1% at 5 years, in the ACOSOG Z0011 and IBCSG 23-01 trials, respectively) compared to the positive non-sentinel lymph nodes rates (27.3% and 13%, respectively) [6,38].

De-escalation in breast cancer treatment refers to the strategy of reducing the intensity or extent of treatment without compromising the efficacy of cancer control. This approach is often implemented to minimize the adverse effects and morbidity associated with more aggressive treatments. The ongoing trend towards therapeutic de-escalation raises critical questions: while reducing the extent of surgical interventions can lessen patient morbidity and improve quality of life, it is crucial to confirm survival data on therapeutic de-escalation in an era of increasing cancer incidence and mortality [39]. Therefore, the difficulty in clinical practice is deciding whether SLNB is sufficient for clinically N0 patients that are pN1 on SLNB. In this population, the rates of pN2/pN3 range between 3.5% to 16% [40], and are particularly low (2.2%) in the ER+/HER2− post-menopausal population [41]. This additional risk of understaging can be evaluated with different clinical tools. Two algorithms are well known predict additional axillary nodal involvement after SLNB: the M.D. Anderson [42] and Memorial Sloan-Kettering [43] nomogram. These algorithms may be useful in clinical practice to decide not only ALND completion but also adjuvant treatments such as radiotherapy [44]. Similarly, a more recent algorithm was published in 2025 for cN0 patients with positive SLNB, and effectively determined the risk of pN2-pN3 involvement in this population with an area under the curve of 0.78 in external validation [45].

Current research also identifies patients in whom axillary staging could be avoided. Indeed, the recent publication of the randomized SOUND [46] and INSEMA [47] trials suggest that omission of axillary surgical staging in early invasive breast cancer is non-inferior to the current SLNB. Therefore, the American Society of Clinical Oncology recently published updated guidelines (2025) that state “clinicians should not recommend routine SLNB in select patients who are postmenopausal and ≥50 years of age and with negative findings on preoperative axillary ultrasound for grade 1–2, small (≤2 cm), hormone receptor-positive, human epidermal growth factor receptor 2-negative breast cancer and who undergo breast-conserving therapy” [48]. The European Society of Medical Oncology has not yet changed its guidelines, which were published in 2024 [2]. More prospective data on this question is expected the coming years, with the publication of the ongoing European BOOG 2013-08 [49], Korean NAUTILIUS [50] and Chinese SOAPET [51] trials. Within this context, a cohort analysis of the SEER database by Wang et al. evaluated the safety of omitting surgical axillary staging in patients with T1 breast cancer treated with breast-conserving surgery. Their findings indicated that, while the 5-year breast cancer-specific survival rates were slightly higher in the staging group compared to the non-staging group, certain subgroups did not benefit significantly from surgical axillary staging. Specifically, patients aged 50–79 years, those with tumors smaller than 1 cm, histological grade I disease, or favorable histological types (such as tubular, mucinous, and papillary), did not show a significant survival benefit from axillary staging [52].

This study’s sample size and statistical methodology, including propensity score matching, strengthen the validity of these findings. By employing propensity score matching, we aimed to minimize selection and indication bias and provide a more accurate comparison of survival outcomes between patients undergoing ALND and those undergoing SLNB. The population is representative of the general breast cancer population, given the distribution of key characteristics such as HER2, ER, and PR status, as well as age distribution. Still, the SEER database and propensity-score usage may have incomplete information on certain variables, adding omitted variable biases. In addition, the methodology for SLNB/ALND classification is not as reliable as the precise surgical information on the axillary procedure, which is not available in the SEER database. It is important to acknowledge that retrospective cohort analyses possess a higher degree of bias compared to level 1 data derived from randomized clinical trials and are not a substitute for them. Nevertheless, the results presented herein are complementary, as they corroborate level 1 data with real-world evidence, providing insight into routine clinical practice, temporal changes, and potentially offering information regarding populations that may not be represented in clinical trials [53]. Finally, another important limitation was the lack of detailed information on adjuvant therapies, both systemic and radiotherapy. Indeed, the SEER database only records whether systemic treatment was prescribed or radiotherapy was delivered. There was no distinction between chemotherapy, endocrine therapy, and targeted therapies, and no detail about what specific protocol was used or the duration. Similarly, it was only recorded whether radiotherapy was delivered, without specification on the regional fields treated. This imbalance could have influenced survival outcomes in our cohort, particularly in the pN2–3 group, where surgical clearance might appear beneficial mainly when nodal radiotherapy is omitted. Therefore, it was not possible to adjust for these critical variables, and part of the observed survival differences between ALND and SLNB may reflect variations in radiotherapy and systemic treatment use rather than the direct therapeutic effect of surgery.

## 5. Conclusions

This study is an in-depth multivariable survival analysis in a very large SEER cohort, comparing SLNB versus ALND in a localized breast cancer population. Our findings indicate that the survival outcomes between SLNB and ALND vary significantly across different N categories, suggesting that the staging benefit decreases with the nodal involvement; conversely, the therapeutic benefit increases as the nodal involvement. This work suggests that the intersection of those benefits (i.e., where ALND may be more useful than SLNB) is between pN1 and pN2 categories. Today, the cut-off point for omitting complementary ALND after a positive SLNB is usually two positive lymph nodes, as no level 1 data safely demonstrate that ALND can be avoided from three positive lymph nodes. Within the context of surgical de-escalation, future research may investigate whether this subgroup (~9% of node-positive patients in this cohort) may also avoid ALND.

## Figures and Tables

**Figure 2 jcm-14-06314-f002:**
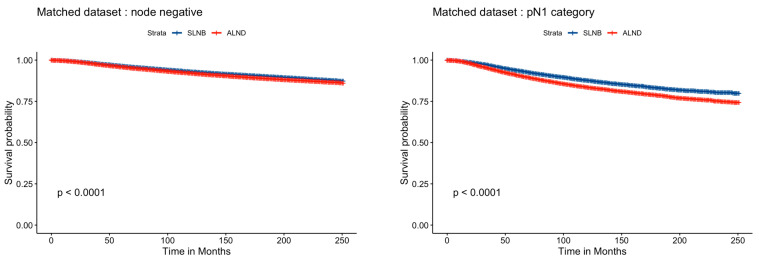
Multivariable propensity-based matched population analysis of survival for pN0 and pN1 patients.

**Table 1 jcm-14-06314-t001:** Clinical, pathological, and treatment characteristics.

	SLNB (N = 524,752)	ALND (N = 300,488)	Total (N = 825,240)	*p*-Value
**Age group (years)**				
<50	93,366 (17.8%)	74,206 (24.7%)	167,572 (20.3%)	<0.001
50–74	343,014 (65.4%)	178,969 (59.6%)	521,983 (63.3%)	
≥75	88,372 (16.8%)	47,313 (15.7%)	135,685 (16.4%)	
**Year of diagnosis**				
Mean (SD)	2011.758 (5.547)	2007.423 (5.801)	2010.180 (6.014)	<0.001
**T category**				
0	87 (0.0%)	373 (0.1%)	460 (0.1%)	<0.001
1	404,953 (77.2%)	170,660 (56.8%)	575,613 (69.8%)	
2	119,712 (22.8%)	129,455 (43.1%)	249,167 (30.2%)	
**N category**				
0	464,573 (88.5%)	138,685 (46.2%)	603,258 (73.1%)	<0.001
1	60,179 (11.5%)	108,624 (36.1%)	168,803 (20.5%)	
2	0 (0.0%)	38,220 (12.7%)	38,220 (4.6%)	
3	0 (0.0%)	14,959 (5.0%)	14,959 (1.8%)	
**Histological subtype**				
Ductal	394,857 (75.2%)	226,400 (75.3%)	621,257 (75.3%)	<0.001
Lobular	46,933 (8.9%)	25,207 (8.4%)	72,140 (8.7%)	
Mixed	50,110 (9.5%)	32,369 (10.8%)	82,479 (10.0%)	
Other	32,852 (6.3%)	16,512 (5.5%)	49,364 (6.0%)	
**Tumor grade**				
1	153,264 (29.2%)	53,773 (17.9%)	207,037 (25.1%)	<0.001
2	241,648 (46.0%)	132,208 (44.0%)	373,856 (45.3%)	
3	129,840 (24.7%)	114,507 (38.1%)	244,347 (29.6%)	
**Estrogen receptors**				
Negative	69,404 (13.2%)	59,127 (19.7%)	128,531 (15.6%)	<0.001
Positive	455,348 (86.8%)	241,361 (80.3%)	696,709 (84.4%)	
**Progesterone receptors**				
Negative	123,001 (23.4%)	92,028 (30.6%)	215,029 (26.1%)	<0.001
Positive	401,751 (76.6%)	208,460 (69.4%)	610,211 (73.9%)	
**HER2**				
Negative	465,878 (88.8%)	251,937 (83.8%)	717,815 (87.0%)	<0.001
Positive	58,874 (11.2%)	48,551 (16.2%)	107,425 (13.0%)	
**Breast-conserving surgery**				
Negative	161,397 (30.8%)	170,594 (56.8%)	331,991 (40.2%)	<0.001
Positive	363,355 (69.2%)	129,894 (43.2%)	493,249 (59.8%)	
**Systemic treatment (adjuvant)**				
Negative	152,934 (29.1%)	95,921 (31.9%)	248,855 (30.2%)	<0.001
Positive	371,818 (70.9%)	204,567 (68.1%)	576,385 (69.8%)	
**Radiotherapy**				
Negative	228,631 (43.6%)	162,706 (54.1%)	391,337 (47.4%)	<0.001
Positive	296,121 (56.4%)	137,782 (45.9%)	433,903 (52.6%)	
**Time between diagnosis and treatment (months)**				
Mean (SD)	1.087 (1.021)	0.853 (1.025)	1.002 (1.029)	<0.001
**Number of removed lymph nodes**				
Mean (SD)	2.368 (1.252)	13.382 (6.435)	6.379 (6.645)	<0.001
Range	1–5	4–82	1–82	
**Number of positive lymph nodes**				
Mean (SD)	0.142 (0.430)	2.068 (3.845)	0.843 (2.522)	<0.001
Range	0–3	0–73	0–73	

SLNB = sentinel lymph node biopsy; ALND = axillary lymph node dissection; SD = standard deviation; HER2 = human epidermal growth-factor receptor 2.

**Table 2 jcm-14-06314-t002:** Multivariable propensity score-based cox model survival analysis for N0-N1 patients.

Matched Population	Hazard Ratio (for ALND)	95%CI	*p*-Value
pN0	1.16	1.12–1.2	<0.001
pN1	1.38	1.3–1.46	<0.001

**Table 3 jcm-14-06314-t003:** Cox model survival analysis for N2-N3 patients.

Parameter	Hazard Ratio	95%CI	*p*-Value
Removed lymph nodes	0. 957	0.955–0.960	<0.001
Positive-to-removed lymph nodes ratio	3.450	2.993–3.976	<0.001

## Data Availability

The SEER database is publicly available. R scripts are available upon reasonable request.

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
