# Peer review of "Is There a Therapeutic Benefit of Axillary Surgery in Non-Metastatic Breast Cancer? A SEER Cohort Database Study"

_jcm, 2025, doi:10.3390/jcm14176314_

Round 1

Reviewer 1 Report

Comments and Suggestions for Authors

To Authors

Congratulations on this well written and done study.

I fully support your results from the analysis in the pN0-1 category.

Regarding your analysis of the pN2-3 group I believe this can be confounded by differences in adjuvant therapy given.  Most patients in the SLND received radiotherapy and most of the patients in the ALND did not receive radiotherapy.  Surgical reductio of tumor burden is more important if radiotherapy is not given. Difference in radiotherapy should be mentioned in your discussion.

Furthermore, small differences in adjuvant treatment like less adjuvant treatment in the ALND can impact survival, even if is only 3% less adjuvant treatment in the ALND. It would also be reasonable to think that a higher proportion in the ALND received adjuvant therapy compared to the SLND because of a more severe disease. Some might have chosen more surgery (ALND) and then omitted adjuvant treatment because of cost.

Breast cancer is a systemic disease. Patients with locally advanced disease have the highest risk of developing distant metastasis and probably have the highest burden of systemic disease. How can extensive surgery reduce the systemic proportion of the disease? My point is that I would have been a bit mor careful in supporting ALND in the pN2-3 group.

I am impressed on how you try to get closer to the truth whit SEER and statistical methods.

Figure 2. Please add Time definition (Time in Months).

Author Response

Dear Reviewer 1,

We sincerely thank you for your encouraging feedback and your thoughtful comments. The issues you raised helped us to strengthen the manuscript and we fully agree with you. Please find below our detailed responses to each of your remarks.

Comment 1: Regarding your analysis of the pN2-3 group I believe this can be confounded by differences in adjuvant therapy given.  Most patients in the SLND received radiotherapy and most of the patients in the ALND did not receive radiotherapy.  Surgical reduction of tumor burden is more important if radiotherapy is not given. Difference in radiotherapy should be mentioned in your discussion.

Response 1: We agree with this important comment. Radiotherapy is a major confounder, as SLNB patients are more likely to receive regional nodal irradiation, while ALND patients often rely on surgery alone for axillary control. We have therefore revised the discussion in two paragraphs (§4 page 8 in the tracked changes version) to explicitly state that differences in radiotherapy use could partly explain the observed survival differences between the two surgery groups in node-positive patients, and we emphasize that SEER does not provide details on irradiated fields, limiting our ability to adjust for this key factor in another paragraph (last paragraph of the discussion, page 10 of the tracked changes version).

Comment 2: Furthermore, small differences in adjuvant treatment like less adjuvant treatment in the ALND can impact survival, even if is only 3% less adjuvant treatment in the ALND. It would also be reasonable to think that a higher proportion in the ALND received adjuvant therapy compared to the SLND because of a more severe disease. Some might have chosen more surgery (ALND) and then omitted adjuvant treatment because of cost.

Response 2: similarly to the previous comment, we also fully agree with the reviewer. Indeed, breast cancer survival is primarily determined by systemic treatment and therefore even minor imbalances in systemic adjuvant therapy between groups can significantly impact survival. We have revised the discussion (§2 of the discussion, page 8 in the tracked changes version, and more importantly in the last paragraph of the discussion, page 10 of the tracked changes version) to acknowledge that part of the survival difference may reflect unequal systemic therapy administration rather than a direct surgical effect, and this analysis has a major limitation as the SEER database does not provide sufficient details about systemic therapies to allow precise analysis.

Comment 3: Breast cancer is a systemic disease. Patients with locally advanced disease have the highest risk of developing distant metastasis and probably have the highest burden of systemic disease. How can extensive surgery reduce the systemic proportion of the disease? My point is that I would have been a bit mor careful in supporting ALND in the pN2-3 group.

Response 3: we completely agree. As stated above, breast cancer is fundamentally a systemic disease, and survival is determined mainly by tumor biology and systemic therapies. While our results show an association between the extent of axillary clearance and survival in pN2–3, this should not be interpreted as proof of a direct therapeutic effect of ALND. We have moderated our interpretation accordingly and revised the discussion (§1 §2 and §4, page 8 in the tracked changes version) to emphasize that the apparent benefit of ALND may reflect residual confounding (radiotherapy or systemic therapy differences) rather than a true systemic effect of surgery.

Comment 4: Figure 2. Please add Time definition (Time in Months)

Response 4: We added the time definition in Figure 2.

Sincerely, 

The corresponding author

Reviewer 2 Report

Comments and Suggestions for Authors

appreciate authors effort to use a large database to identify the impact of extent of axillary surgery on survival. However, the study cohort spans over 20 years where both surgical and non surgical adjuvant treatment of breast cancer has changed a lot. Authors have acknowledged the limitation of classification into SLNB and ALND groups.  Other queries are as follows

  1. It is not clear from the methodology, how did you account for patients who had completion axillary node clearance after initial SLNB which shown positive LNs
  2. There is no data on adjuvant Nodal Radiotherapy which can have impact on longer term survival

Author Response

Dear Reviewer,

We sincerely thank you for your constructive comments and valuable suggestions, which allowed us to clarify our methodology and raised an important limitation which we discussed more explicitly. Please find below our detailed responses to each of your remarks.

Comment 1: It is not clear from the methodology, how did you account for patients who had completion axillary node clearance after initial SLNB which shown positive LNs

Response 1: thank you for pointing this out. The SEER database does not include this parameter, and it is not possible to accurately discriminate upfront ALND and ALND completion after positive SLNB. We have clarified this limitation in methodology (page 2 and 3 of the tracked changes version) to underline this “The exact type of axillary surgical procedure is not explicitly reported in the SEER Database (SLNB or ALND), nor whether it was upfront ALND or completion after initial SLNB.” and “However, this approach did not allow us to discriminate upfront ALND and SLNB followed by ALND. “

Comment 2: There is no data on adjuvant Nodal Radiotherapy which can have impact on longer term survival

Response 2: We fully agree and thank the reviewer for pointing this out. Nodal radiotherapy is a key determinant of outcomes in node-positive breast cancer. SEER records only whether radiotherapy was given, without specifying the fields treated. We have revised the discussion (last paragraph, page 10 and §2, page 8 of the tracked changes version) to acknowledge this limitation explicitly, and to note that variations in nodal radiotherapy delivery could have significantly influenced survival outcomes in our study.

Sincerely, 

The corresponding author

Round 2

Reviewer 2 Report

Comments and Suggestions for Authors

Thank you for for explaining the limitations of the study in detail which is important for the readers when interpreting the outcome